

# Ensembles of decision trees and gradient-based learning for employee turnover rate prediction

Chunyang Zhang[1] and Wenjing Han[2]

[1] School of Labor Economics, Capital University of Economics and Business, Beijing, China
[2] School of Government, Beijing Normal University, Beijing, China

## ABSTRACT

Employee turnover has a negative impact on business profitability. To tackle this issue, we can utilize computational advancements to forecast attrition and minimize expenses. We employed an HR Analytics dataset to investigate the feasibility of using these predictive models in decision support systems. We developed an ensemble of gradient-based decision trees that accurately predicted employee turnover and performed better than other sophisticated techniques. This approach demonstrates exceptional performance in handling structured and imbalanced data, effectively capturing intricate patterns. Gradient-based decision trees provide scalable solutions that effectively balance predictive accuracy and computational efficiency, making them well-suited for strategic business analysis. The importance of our findings lies in their ability to offer dependable insights for making well-informed decisions in business settings.

## INTRODUCTION

The employee turnover rate is a key business metric that quantifies the rate at which employees leave a company within a specific period, usually expressed as a percentage (*Woods, 2015*; *Chiat & Panatik, 2019*; *Bolt, Winterton & Cafferkey, 2022*). This rate is a ratio between the number of employees who have left the company and the average number of people who are working during the same period (*Hurley & Estelami, 2007*; *Li et al., 2016*). A high turnover rate can indicate various underlying issues, such as job dissatisfaction, poor workplace culture, inadequate compensation, or lack of career advancement opportunities (*Ross & Zander, 1957*; *Raza et al., 2022*). Conversely, a low turnover rate often reflects a positive working environment, job satisfaction, and effective employee retention strategies (*Ross & Zander, 1957*; *Raza et al., 2022*). Furthermore, employee turnover rates can be influenced by cultural (*Yao & Wang, 2006*; *Wu, Rafiq & Chin, 2017*) and national factors (*Peretz & Fried, 2012*; *Kwakye, 2018*; *Ilmi et al., 2019*). Different regions have distinct workplace norms, job expectations, and employee engagement practices that can significantly impact retention and turnover (*Bolt, Winterton & Cafferkey, 2022*). In cultures where long-term job stability is highly valued, turnover rates might be lower (*Hamermesh, Hassink & van Ours, 1996*). On the other hand,

Corresponding author
Chunyang Zhang,
zhangcueb@126.com

environments that emphasize career progression and individual achievement may exhibit higher turnover rates as employees pursue better opportunities for advancement (*Dipietro & Condly, 2007*). Additionally, the legal and economic framework of a country also significantly affects turnover, with stricter labor laws leading to lower turnover rates and more flexible labor markets experiencing higher rates (*Guthrie, 2000*; *Zhang, 2016*). Therefore, understanding the cultural and national context is essential for multinational companies aiming to develop effective human resource strategies that meet the diverse expectations and norms of their global workforce (*Bolt, Winterton & Cafferkey, 2022*). Monitoring and analyzing turnover rates helps organizations identify trends, assess the effectiveness of their human resource policies, and implement measures to improve employee engagement and retention, ultimately impacting the organization's performance and stability (*Wulansari, Meilita & Ganesan, 2020*). Attrition rate, another business metric for the same purpose, captures the concept of workforce reduction over time (*Peng, 2022*). While the 'turnover rate' specifically refers to the rate at which employees leave and are replaced within an organization, the 'attrition rate' is more broadly focused on the net reduction of the workforce (*Pallathadka et al., 2022*). In case data for calculating the 'turnover rate' are not available or missing, the 'attrition rate' can be employed as a more generalized way to discuss workforce reduction without needing to quantify the inflow and outflow of employees precisely (*Speer et al., 2019*). In fact, datasets for 'turnover rate' calculation are usually smaller than those for 'attrition rate' calculation since companies usually need to spend a bigger budget for collecting 'turnover rate' data rather than for collecting 'attrition rate' data. The 'attrition rate' data are usually stored in the company's databases and can be easily retrieved for further analysis. In other words, in the prediction of the turnover rate, 'attrition rate' data can be used for the same purpose, and even better for a broader scale analysis of job quitting factors.

Attrition has complex consequences for any organization, impacting it in both beneficial and detrimental ways (*Peng, 2022*). On the plus side, attrition can streamline the workforce by naturally eliminating less effective or disengaged employees. This process opens opportunities for fresh talent to enter, bringing innovative ideas and new energy, which can boost organizational productivity and better align the workforce with evolving market demands (*Reeder, Henderson & Sullivan, 1982*). Attrition can also help in financial management by eliminating the need for forced redundancies, thus allowing a more natural adjustment of staff levels in line with business cycles or strategic changes (*Bellucci, 2014*). On the downside, attrition can be disruptive, especially when it results in the loss of critical staff. Such departures can lead to gaps in knowledge and expertise, disrupting workflows and affecting the quality and continuity of business operations (*Marchiondo, Cortina & Kabat-Farr, 2018*). This is particularly problematic in industries where skilled professionals are rare and difficult to replace, heightening operational vulnerabilities and increasing the workload on remaining employees (*Hughes & Trafimow, 2011*). A high rate of attrition might also reflect deeper problems within the organization, such as poor management, inadequate compensation, or insufficient opportunities for career progression (*Li et al., 2023*). A recent survey has pointed out that nearly one-third of new employees tend to resign within their first 6 months of employment (*Peng, 2022*).

According to the Job Openings and Labor Turnover Survey (JOLTS), the United States sees around 4.5 million job resignations monthly. Apollo Technical's findings suggest that the attrition rate hovers around 19% across different sectors. In contrast, the Bureau of Labor Statistics reports a spike in the attrition rate, reaching above 57% in 2021 within the U.S. (*Raza et al., 2022*). For a business to operate effectively, maintaining a high employee retention rate, ideally around 90%, is crucial, while keeping the attrition rate under 10%. It's essential for companies not only to monitor these rates but also to understand the reasons behind employee departures. Implementing strategies to address these reasons can lead to improved employee satisfaction and loyalty, thereby enhancing overall organizational stability and performance (*Raza et al., 2022*).

Recent years have seen an outgrowth of artificial intelligence in the age of information explosion (*Berman, 2008*). The availability of enormous sources of data has motivated the development of numerous computational frameworks to tackle problems in diverse fields of life (*Ourmazd, 2020*). Machine learning-based frameworks, therefore, have been created to address existing issues or improve limitations of present technologies (*Shinde & Shah, 2018*). As an essential sub-domain of artificial intelligence, machine learning models are trained to assist decision-making processes (*Xu, Li & Donta, 2024*). In today's business, most companies are supported by machine learning-based analytics tools to enhance working efficiency. The complexity of these tools depends on the purpose of use (*Bose & Mahapatra, 2001*; *Song, Cao & Zhang, 2018*; *Leow, Nguyen & Chua, 2021*). *Qutub et al. (2021)* developed a computational model for employee attrition prediction. To find the most suitable model, they used multiple machine learning algorithms in combination with data retrieved from the database on employees at IBM. *Habous, Nfaoui & Oubenaalla (2021)* constructed various prediction frameworks using random forest, AdaBoost, gradient boosting, decision tree, and logistic regression to predict employee attrition. Their findings indicated that the model developed with logistic regression outperformed the others in terms of accuracy. *Najafi-Zangeneh et al. (2021)* also employed logistic regression for modeling with an accuracy of 81%. In their work, max-out feature selection was used to reduce the input vector's dimension. *Pratt, Boudhane & Cakula (2021)* performed an extensive survey on the performance of a wide range of learning algorithms to suggest the most adaptive ones for developing models that can effectively predict attrition rate. *Sadana & Munnuru (2022)* conducted a study to address the attrition issues in IT firms using machine learning approaches. Their results raise several problems in managing activity and working environment. *Kaya & Korkmaz (2021)* trained a series of machine learning models for measuring the staff turnover rate. Since the dataset was not balanced, class rebalancing techniques were used, and feature selection and bootstrapping were employed to improve prediction efficiency. Access to such insights has empowered companies to swiftly implement measures aimed at retaining employees who were unhappy with various aspects of their job, such as the workplace environment, work-life balance, chances for promotion, and other significant elements. These innovative methods contribute to boosting employee contentment by providing a more precise forecast of turnover trends. However, most existing approaches in this field are based on classical machine learning algorithms (*Wang, Nguyen & Nguyen, 2020*). Although these methods yield promising

results, there is significant room for improvement. Exploring more efficient computational methods is critical not only for prediction accuracy, but also for adapting to the increasing complexity and scale of data in organizational contexts.

In an effort to address this problem, we aim to develop a more effective computational model using ensembles of multiple decision trees under gradient-based learning. Ensemble learning, which combines multiple models to improve overall performance, has proven highly effective across various domains (*Pham et al., 2019*; *Nguyen et al., 2022*; *Wang, Chukova & Nguyen, 2023a*) by leveraging the strengths of diverse algorithms to achieve superior predictive accuracy and robustness. Our approach is specifically designed to be well-adapted to tabular data, which is commonly used in many business and industrial applications. For our study, we utilized the HR Analytics dataset to train and assess the performance of our model, ensuring its applicability in real-world scenarios involving employee turnover and retention predictions. Given that most existing prediction models in this field rely on classical machine learning techniques, our model is benchmarked against those built on seven widely-used machine learning algorithms: eXtreme Gradient Boosting (XGB) (*Chen & Guestrin, 2016*), AdaBoost (AB) (*Friedman, 2002*, *2001*), random forest (RF) (*Breiman, 2001*), logistic regression (LR) (*LaValley, 2008*), decision tree (DT) (*de Ville, 2013*), $k$-nearest neighbors (k-NN) (*Kramer, 2013*), and support vector machine (SVM) (*Suthaharan, 2016*), and two advanced Transformer-based deep learning algorithms: TabTransformer (*Huang et al., 2020*) and FT-Transformer (*Gorishniy et al., 2021*). This comprehensive evaluation aims to demonstrate the enhanced accuracy and generalization capabilities of our ensemble approach for predictive modeling in HR analytics and beyond.

## EXPERIMENT DESIGN

Our study focuses on developing a prediction model based on ensembles of decision trees and gradient-based learning to predict employee turnover rate. To develop our model, we used the HR Analytics dataset. Detailed information on the dataset and how the data were sampled is provided in the next section. Besides our proposed model, we also developed seven conventional machine learning models and two deep learning models to fairly assess the model performance. All models were trained, optimized, and tested with the same datasets. For machine learning models, the training and validation sets were merged to create a new training set. The training set was then used for model optimization with 5-fold cross-validation. For deep learning models, the optimal models were obtained based on the validation loss. The deep learning models were trained over 100 epochs, with a learning rate of 0.001 and optimized by AdamOptimizer.

## DATASET

We used the HR Analytics dataset from *Sisodia, Vishwakarma & Pujahari (2017)* to develop our prediction model. The original dataset contained 15,000 samples. After removing unqualified samples, we obtained a refined dataset of 14,499 samples. This refined dataset was randomly split into two sets: training data (80%) and test data (20%). From the training data, we allocated 15% to create a validation set, resulting in 1,800

samples for validation and 10,199 samples for training; the test set consisted of 3,000 samples. Samples are defined by six continuous variables, including: *satisfaction_level*, *last_evaluation*, *number_project*, *average_montly_hours*, *time_spend_company*, *work_accident*, and two categorical variables: '*department*' and '*salary*'. Detailed information about these datasets is provided in Table 1. Figure 1 shows the proportion of employees who left, segmented by department and by salary level. The plot reveals that the Human Resource (HR) department has a higher proportion of resignations compared to others. In contrast, Management and Research and Development (R&D) departments have a lower proportion of resigned employees. Additionally, workers receiving low salaries tend to leave their jobs more frequently than those earning medium or high salaries.

Figure 2 displays the correlation among the variables in the dataset, using Pearson's correlation coefficient ($r$) as the metric. Most variables exhibit an absolute Pearson's $r$ value of less than 0.5, with the range extending from approximately −0.4 to 0.4. This mild correlation indicates that no variable strongly influences another, which suggests a diverse set of features for analysis. Following a comprehensive correlation analysis of all variables, it was decided to retain all of them for the modeling process to ensure a holistic representation of the factors affecting the outcome.

## METHODOLOGY

### Method overview

Figure 3 summarizes all the stages in our study. First, after refining the dataset, a test set accounting for 20% of the total samples is created using random sampling. The remaining data is then used to create a training set and a validation set with a ratio of 85:15. The class distribution of these three datasets is kept unchanged using stratified sampling. The training set is used for model construction, while the validation set is responsible for finding the best hyperparameters for the machine learning models and the stopping epoch for deep learning models. Once model optimization is completed, all hyperparameter-refitted models from each algorithm are gathered for testing using the test dataset.

### Gradient-based decision tree

Decision trees are powerful algorithm used for both classification and regression tasks, characterized by their hierarchical structure that facilitates decision-making. Their interpretability and flexibility make them popular in various fields. Gradient-based decision tree algorithm enhances traditional tree-based algorithms by incorporating the principles of gradient boosting, leading to a more dynamic and powerful model often used in ensemble learning methods. These trees operate on the principle of optimizing a specified loss function, which can be tailored for various types of predictive modeling tasks, including regression and classification. Unlike traditional decision trees provide a simple and interpretable model, The gradient-based decision tree involves constructing weak learners, specifically decision trees, in a stage-wise fashion where each tree is built to correct the errors of its predecessors, effectively implementing a gradient descent-like approach to minimize the overall model error (*Marton et al., 2023*). The gradient-

| Table 1 Information on training, validation, and test sets. | | | |
|---|---|---|---|
| **Data** | **Number of samples** | | |
| | **Positive** | **Negative** | **Total** |
| Training | 2,442 | 7,757 | 10,199 |
| Validation | 428 | 1,372 | 1,800 |
| Test | 701 | 2,299 | 3,000 |
| **Total** | **3,571** | **11,428** | **14,999** |

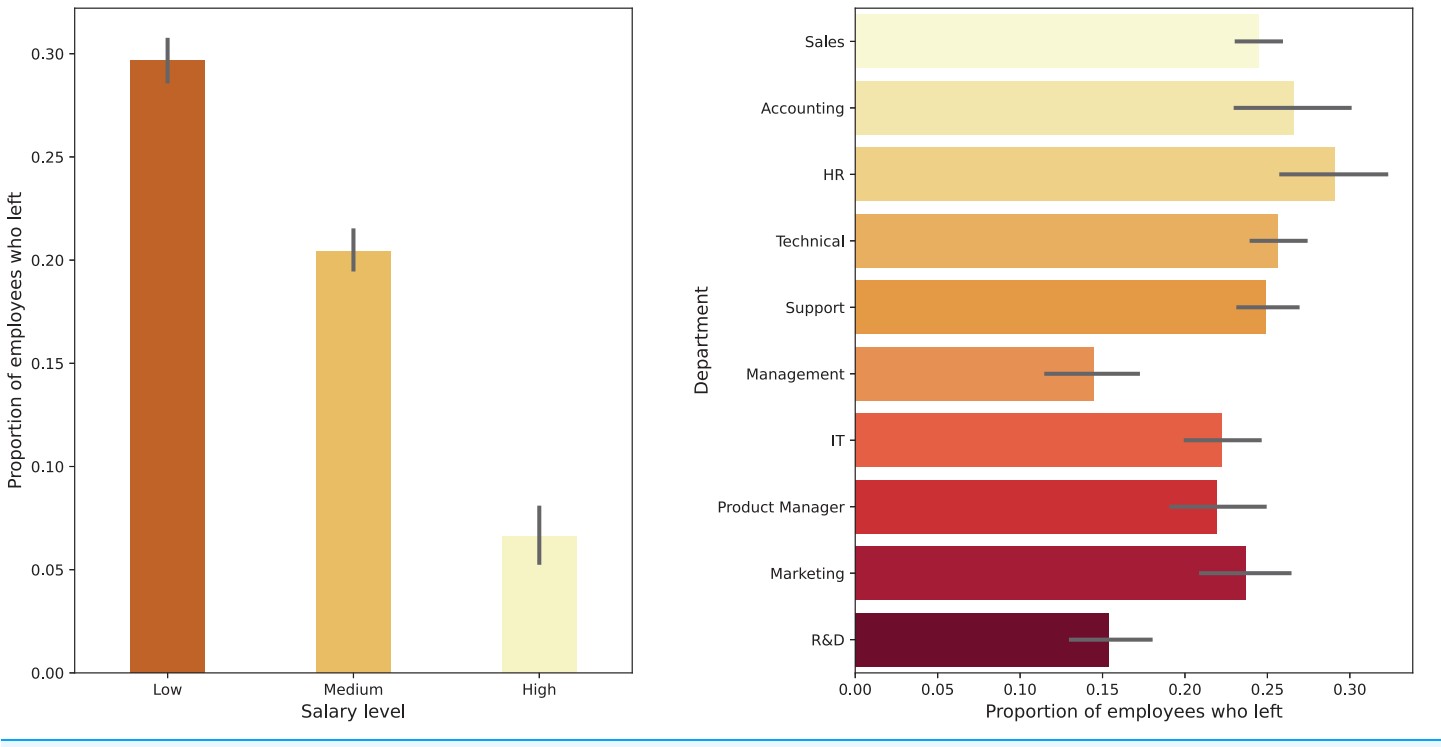

**Figure 1 Proportion of employees who left, segmented by department.**   

algorithm to adaptively refine the model by focusing on the difficult parts of the prediction task, thereby gradually enhancing its predictive accuracy. Gradient-based decision trees are known for their flexibility, allowing them to tackle different data and problem types. Additionally, they possess a strong predictive power with the capability of handling complex, non-linear relationships. Moreover, they include regularization features that help in controlling overfitting, making them robust against training on noisy data. eXtreme Gradient Boosting (*Chen & Guestrin, 2016*), LightGBM (*Ke et al., 2017*), and CatBoost (*Prokhorenkova et al., 2018*) are typical learning algorithm developed based on gradient-based decision trees. A normal decision tree $t$ with depth $d$ is defined as:

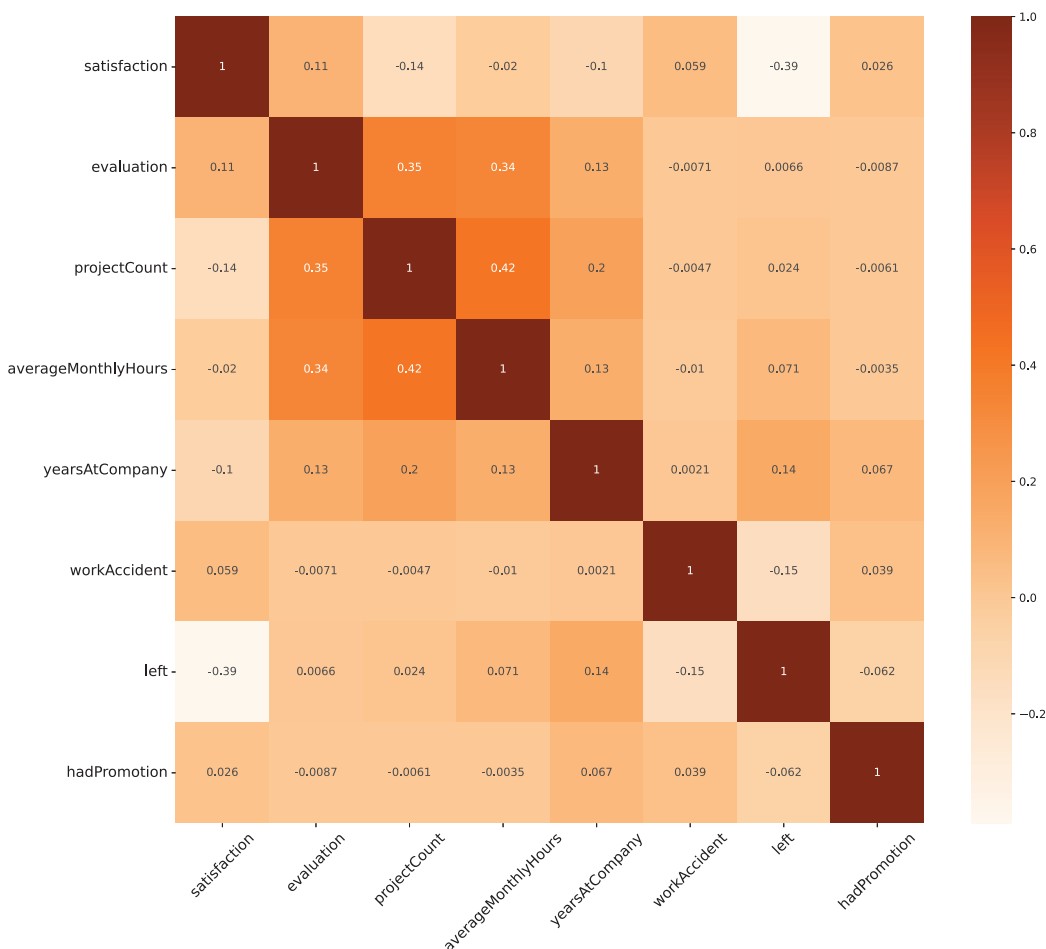

**Figure 2 Heat map for correlation analysis on all variables.**

$$t(x|\lambda, \tau, \iota) = \sum_{l=0}^{2^d-1} \lambda_l \mathbb{L}(x|\lambda, \tau, \iota), \tag{1}$$

where $\mathbb{L}$ is a function that determines whether a sample $x \in \mathbb{R}^n$ belongs to a leaf $l$, $\lambda \in C^{2^d}$ indicates the class of the sample associated with a single leaf node, $\tau \in \mathbb{R}^{2^d-1}$ represents the threshold for splitting, and $\iota \in \mathbb{N}^{2^d-1}$ denotes the index of features for each internal node. Given a gradient-based decision tree $g$ at depth $j$ with leaf node $l$, the function of tree is rewritten as:

$$g(x|\lambda, T, I) = \sum_{l=0}^{2^d-1} \lambda_l \mathbb{L}(x|\lambda, T, I), \tag{2}$$

$$\mathbb{L}(x|\lambda, T, I) = \prod_{j=1}^{d} (1 - p(l,j))\mathbb{S}(x|I_{i(l,j)}, T_{i(l,j)}) + p(l,j)(1 - \mathbb{S}(x|I_{i(l,j)}, T_{i(l,j)})), \tag{3}$$

where $p$ is binary probability with $p = 0$ or 1 when the left or right branch is taken,

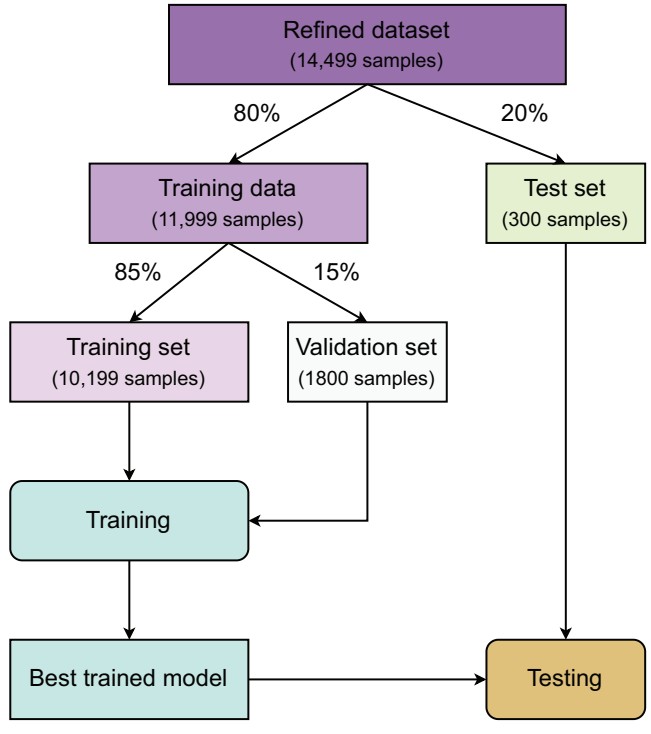

**Figure 3** **Method overview.**

$T \in \mathbb{R}^{(2^d-1) \times n}$ and $I \in \mathbb{R}^{(2^d-1) \times n}$ are matrixized transformations of the threshold for splitting and index of features, respectively, and $\mathbb{S}(x)$ represents for the logistic function.

While traditional decision trees provide a simple and interpretable model, GBDT enhances predictive accuracy through an ensemble approach that sequentially builds trees to correct errors, incorporates regularization, and optimizes performance *via* gradient descent. This makes GBDT a powerful tool in machine learning, particularly for complex datasets where traditional methods may fall short.

## Ensembles of gradient-based decision trees

To empower the gradient-based decision tree, we designed an ensemble learning algorithm leveraging a large number of trees. The ensemble of gradient-based decision tree $G$ is expressed as:

$$G(x|\omega, \mathbf{L}, \mathbf{T}, \mathbf{I}) = \sum_{e=0}^{E} \omega_e g(x|\mathbf{L}_e, \mathbf{T}_e, \mathbf{I}_e), \tag{4}$$

where $E$ and $\omega$ are the number of trees and weight vector, respectively. Having end-to-end learning process driven by gradient descent, our method is expected to address limitations of traditional non-gradient-based tree methods like XGBoost (*Chen & Guestrin, 2016*) or CatBoost (*Prokhorenkova et al., 2018*). Figure 4 visualizes the architecture of ensembles of gradient-based decision trees.

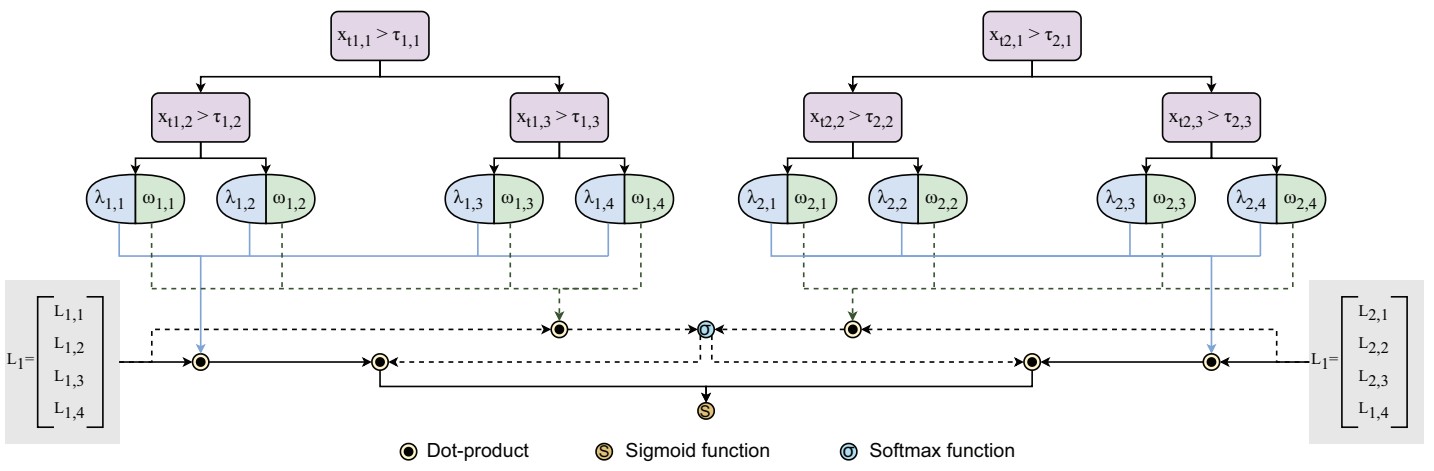

**Figure 4 Architecture of ensembles of gradient-based decision trees.**

# RESULTS AND DISCUSSION

## Model implementation

After completing the model design, we trained and optimized our model using the training and validation sets, respectively. To evaluate our model fairly, we implemented seven machine learning models, including XGB (*Chen & Guestrin, 2016*), AB (*Friedman, 2002, 2001*), RF (*Breiman, 2001*), LR (*LaValley, 2008*), DT (*de Ville, 2013*), *k*-NN (*Kramer, 2013*), and SVM (*Suthaharan, 2016*). Additionally, two advanced Transformer-based deep learning models: TabTransformer (*Huang et al., 2020*) and FT-Transformer (*Gorishniy et al., 2021*) were also implemented for comparison. All models were implemented under the same conditions. Once obtained, these models were evaluated on an independent test set to verify their predictive power.

## Assessment metrics

We utilized several metrics to assess model performance, including area under the receiver operating characteristic curve (AUCROC), area under the precision-recall curve (AUCPR), and accuracy (ACC). Derived from true positive, false positive, true negative, and false negative rates, these metrics are pivotal in machine learning for evaluating classification models. AUCROC gauges a model's ability to discriminate between positive and negative classes, while AUCPR is crucial for analyzing the precision-recall trade-off, particularly in datasets with uneven class distribution. These indicators not only provide a holistic view of model efficacy but are also adept at managing class imbalances, thus facilitating the evaluation process. Their comprehensive nature enables a nuanced analysis of a model's predictive quality, greatly influencing model selection and optimization strategies and offering insights into the classifier's discriminative capability. Additionally, we employed balanced accuracy (BA), Matthew's correlation coefficient (MCC), Cohen's kappa (CK), sensitivity (SN), specificity (SP), precision (PRE), and F1 score (F1) to further enrich our evaluation framework.

**Table 2 Benchmarking results between our proposed method and the others.**

| Method | Metric | | | | | | | | |
|---|---|---|---|---|---|---|---|---|---|
| | ROCAUC | PRAUC | BA | MCC | CK | SN | SP | PRE | F1 |
| XGB | 0.9794 | 0.9605 | 0.9555 | 0.9342 | 0.9158 | 0.9952 | 0.9832 | 0.9483 | 0.9333 |
| AB | 0.9578 | 0.8966 | 0.8420 | 0.7647 | 0.6976 | 0.9865 | 0.9404 | 0.8010 | 0.7515 |
| RF | 0.9621 | 0.8895 | 0.9621 | 0.9170 | 0.9472 | 0.9769 | 0.9261 | 0.9365 | 0.9169 |
| LR | 0.8065 | 0.4908 | 0.5852 | 0.2228 | 0.2496 | 0.9208 | 0.4902 | 0.3308 | 0.2055 |
| DT | 0.9793 | 0.9315 | 0.9793 | 0.9503 | 0.9743 | 0.9843 | 0.9499 | 0.9619 | 0.9502 |
| k-NN | 0.9685 | 0.8873 | 0.9371 | 0.8462 | 0.9272 | 0.9469 | 0.8420 | 0.8825 | 0.8445 |
| SVM | 0.8093 | 0.5555 | 0.5863 | 0.2686 | 0.2126 | 0.9600 | 0.6183 | 0.3164 | 0.2235 |
| TabTransformer | 0.9789 | 0.9290 | 0.9789 | 0.9486 | 0.9743 | 0.9835 | 0.9473 | 0.9606 | 0.9484 |
| FT-Transformer | 0.9795 | 0.9328 | 0.9795 | 0.9512 | 0.9743 | 0.9848 | 0.9513 | 0.9627 | 0.9511 |
| Ours | 0.9892 | 0.9795 | 0.9573 | 0.9333 | 0.9215 | 0.9930 | 0.9758 | 0.9479 | 0.9326 |

## Model benchmarking

Table 2 provides benchmarking results comparing our proposed method with others. The results show that our model is more effective compared to other methods in terms of ROCAUC and PRAUC. Our model achieves a ROCAUC value of 0.9892, followed by the FT-Transformer model, the TabTransformer model, and other conventional machine learning models. Among these conventional machine learning models, the XGB model is the best model, followed by DT, k-NN, RF, AB, and other models. For PRAUC, our model obtains a value of 0.9795. The LR and SVM models have the smallest PRAUC values, at 0.4908 and 0.5555, respectively. Except for the AB and k-NN models, all other models have PRAUC values over 0.9. MCC is a statistical metric used to evaluate the quality of binary classifications. As it accounts for true and false positives and negatives, it is a balanced metric suitable even when the classes are of very different sizes. The MCC value for the FT-Transformer model is 0.9512, which is higher than that for the DT model, the XGB model, and our method, with 0.9503, 0.9342, and 0.9333, respectively. Our method ranks as the fourth best based on MCC, with a threshold of 0.5. In terms of sensitivity, specificity, and precision, our model's performance is only slightly lower than that of the XGB model. Compared to the other models, the FT-Transformer model has the highest F1 score.

## Repeated experiments

To investigate the robustness of our proposed method, we repeated the experiments 20 times with 20 different test sets created by random sampling (Table 3). The results show that the average ROCAUC and PRAUC values vary within small ranges of 0.0043 and 0.0161, respectively. Across 20 runs, the ROCAUC values ranged from 0.96 to 0.98, and the PRAUC values from 0.90 to 0.97 (Fig. 5). Although the PRAUC value exhibits a larger standard deviation compared to other metrics, it remains small and acceptable. The findings of this experiment demonstrate that our proposed method is both effective and

**Table 3 Results of repeated experiments for assessing the robustness of the model.**

| Trial | ROCAUC | PRAUC | BA | MCC | CK | SN | SP | PRE | F1 |
|---|---|---|---|---|---|---|---|---|---|
| 1 | 0.9892 | 0.9795 | 0.9573 | 0.9333 | 0.9215 | 0.9930 | 0.9758 | 0.9479 | 0.9326 |
| 2 | 0.9743 | 0.9121 | 0.9743 | 0.9359 | 0.9705 | 0.9781 | 0.9326 | 0.9512 | 0.9356 |
| 3 | 0.9741 | 0.9226 | 0.9741 | 0.9418 | 0.9659 | 0.9824 | 0.9465 | 0.9561 | 0.9417 |
| 4 | 0.9774 | 0.9328 | 0.9774 | 0.9496 | 0.9699 | 0.9850 | 0.9542 | 0.9620 | 0.9496 |
| 5 | 0.9746 | 0.9226 | 0.9746 | 0.9426 | 0.9663 | 0.9829 | 0.9464 | 0.9562 | 0.9425 |
| 6 | 0.9778 | 0.9320 | 0.9778 | 0.9495 | 0.9710 | 0.9846 | 0.9526 | 0.9617 | 0.9494 |
| 7 | 0.9780 | 0.9302 | 0.9780 | 0.9485 | 0.9723 | 0.9838 | 0.9499 | 0.9610 | 0.9484 |
| 8 | 0.9733 | 0.9248 | 0.9733 | 0.9442 | 0.9605 | 0.9862 | 0.9535 | 0.9570 | 0.9442 |
| 9 | 0.9778 | 0.9182 | 0.9778 | 0.9408 | 0.9781 | 0.9775 | 0.9333 | 0.9552 | 0.9403 |
| 10 | 0.9767 | 0.9224 | 0.9767 | 0.9434 | 0.9717 | 0.9817 | 0.9424 | 0.9568 | 0.9432 |
| 11 | 0.9676 | 0.9137 | 0.9676 | 0.9334 | 0.9523 | 0.9828 | 0.9472 | 0.9497 | 0.9334 |
| 12 | 0.9747 | 0.9023 | 0.9747 | 0.9301 | 0.9760 | 0.9734 | 0.9188 | 0.9465 | 0.9293 |
| 13 | 0.9833 | 0.9407 | 0.9833 | 0.9569 | 0.9825 | 0.9840 | 0.9531 | 0.9676 | 0.9567 |
| 14 | 0.9764 | 0.9182 | 0.9764 | 0.9405 | 0.9734 | 0.9794 | 0.9367 | 0.9547 | 0.9402 |
| 15 | 0.9778 | 0.9323 | 0.9778 | 0.9493 | 0.9716 | 0.9841 | 0.9523 | 0.9619 | 0.9492 |
| 16 | 0.9769 | 0.9266 | 0.9769 | 0.9461 | 0.9704 | 0.9834 | 0.9477 | 0.9589 | 0.9460 |
| 17 | 0.9727 | 0.9228 | 0.9727 | 0.9414 | 0.9616 | 0.9837 | 0.9499 | 0.9557 | 0.9414 |
| 18 | 0.9760 | 0.9120 | 0.9760 | 0.9369 | 0.9741 | 0.9779 | 0.9300 | 0.9515 | 0.9365 |
| 19 | 0.9727 | 0.9080 | 0.9727 | 0.9333 | 0.9666 | 0.9788 | 0.9315 | 0.9487 | 0.9331 |
| 20 | 0.9778 | 0.9376 | 0.9778 | 0.9526 | 0.9689 | 0.9867 | 0.9598 | 0.9643 | 0.9526 |
| Mean | 0.9765 | 0.9256 | 0.9749 | 0.9425 | 0.9673 | 0.9825 | 0.9457 | 0.9562 | 0.9423 |
| Std | 0.0043 | 0.0161 | 0.0052 | 0.0072 | 0.0126 | 0.0042 | 0.0126 | 0.0058 | 0.0074 |

stable. Table 4 presents computed confidence interval (CI) over 20 repeated trails with $\alpha$ values of 0.01, 0.05, and 0.1 correponding to 99%CI, 95%CI, and 90%CI, respectively.

## Limitations and future work

While gradient-based decision trees offer significant advantages in terms of predictive power and flexibility, they also come with limitations related to overfitting, computational demands, and interpretability (*Grinsztajn, Oyallon & Varoquaux, 2022*). Their applicability is strongest in structured data environments, particularly where feature importance and handling of imbalanced datasets are crucial. Understanding these factors is essential for effectively leveraging the performance of machine learning models based on this algorithm.

To further enhance the robustness and generalizability of models using gradient-based decision trees, future work could explore the integration of generative models or advanced resampling methods for tabular data. Generative models could be employed to synthesize realistic data samples, thereby mitigating issues of data scarcity and imbalance (*Wang et al., 2024*). Additionally, advanced resampling techniques designed specifically for

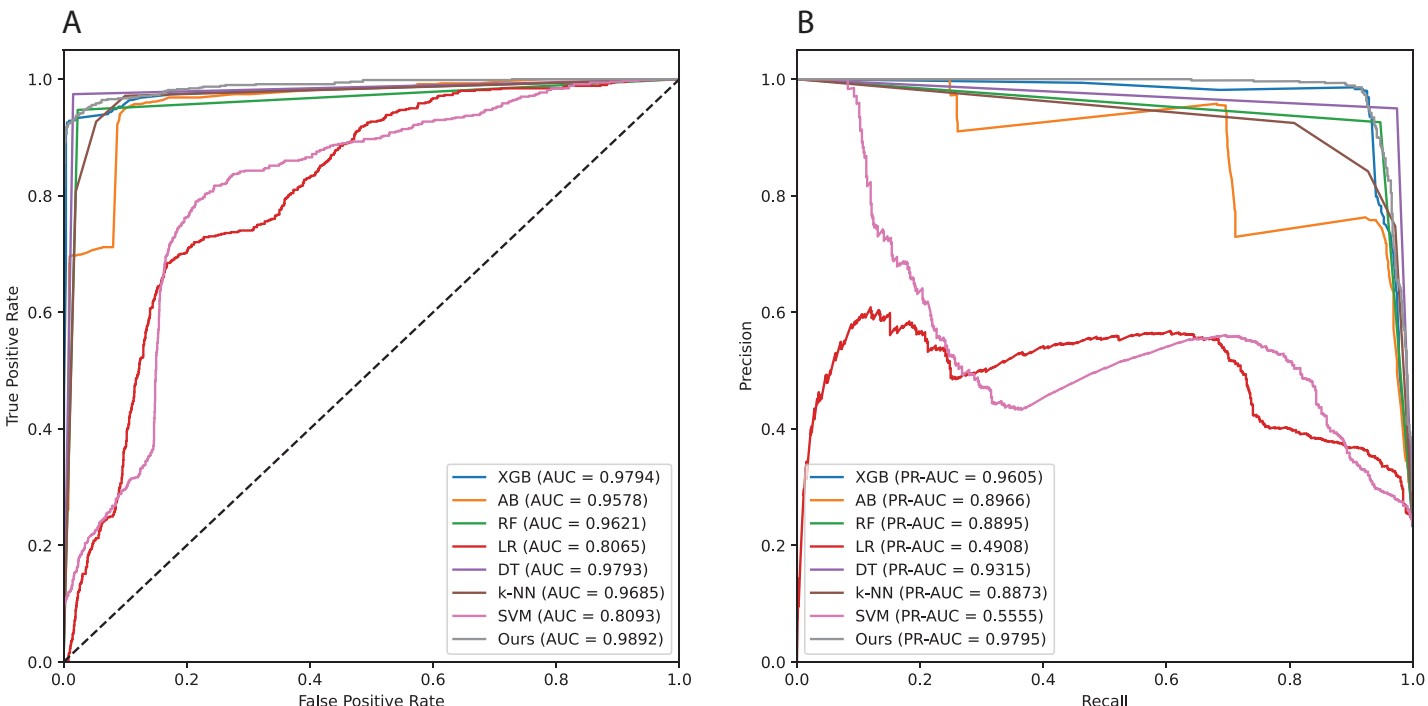

**Figure 5 Area under the curves of all implemented models.** (A) Receiver operating characteristic (ROC) curve. (B) Precision-Recall (PR) curve.

**Table 4 Confidence interval over 20 repeated trials.**

| Metric | Confidence interval | | |
|---|---|---|---|
| | **99%** | **95%** | **90%** |
| ROCAUC | [0.9749–0.9781] | [0.9746–0.9784] | [0.9756–0.9774] |
| PRAUC | [0.9197–0.9315] | [0.9185–0.9327] | [0.9222–0.9290] |
| BA | [0.9730–0.9768] | [0.9726–0.9772] | [0.9738–0.9760] |
| MCC | [0.9399–0.9451] | [0.9393–0.9457] | [0.9410–0.9440] |
| CK | [0.9627–0.9719] | [0.9618–0.9728] | [0.9646–0.9700] |
| SN | [0.9810–0.9840] | [0.9807–0.9843] | [0.9816–0.9834] |
| SP | [0.9411–0.9503] | [0.9402–0.9512] | [0.9430–0.9484] |
| PRE | [0.9541–0.9583] | [0.9537–0.9587] | [0.9550–0.9574] |
| F1 | [0.9396–0.9450] | [0.9391–0.9455] | [0.9407–0.9439] |

tabular data (*Wang, Chukova & Nguyen, 2023b*) could help address class imbalance and improve model performance. Incorporating these approaches could enhance the model's ability to generalize across diverse datasets and further reduce the risks of overfitting, making gradient-based decision trees even more powerful and applicable to a broader range of real-world problems.

## CONCLUSION

In this study, we developed a computational framework to predict employee turnover rate using ensembles of gradient-based decision trees. The results show that our model performs more effectively compared to other methods. While gradient-based decision trees offer the strength of handling various types of data and capturing complex nonlinear relationships, they can also be computationally intensive and sensitive to overfitting, especially with large datasets. Despite these challenges, the flexibility and predictive power of this approach are significant, particularly when enhanced with appropriate regularization techniques. Future improvements could focus on optimizing computational efficiency and further minimizing the risk of overfitting. This method holds potential for addressing a wide range of problems beyond employee turnover, suggesting its applicability in diverse analytical scenarios.

### Funding
The authors received no funding for this work.

### Competing Interests
The authors declare that they have no competing interests.

### Author Contributions
- Chunyang Zhang conceived and designed the experiments, performed the experiments, analyzed the data, performed the computation work, prepared figures and/or tables, authored or reviewed drafts of the article, and approved the final draft.
- Wenjing Han conceived and designed the experiments, performed the experiments, analyzed the data, performed the computation work, prepared figures and/or tables, authored or reviewed drafts of the article, and approved the final draft.

### Data Availability
The dataset and the Python code used in this study are available in the Supplemental File.

The data is available at Kaggle: https://www.kaggle.com/datasets/giripujar/hr-analytics.

### Supplemental Information
Supplemental information for this article can be found online at http://dx.doi.org/10.7717/peerj-cs.2387#supplemental-information.

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
