# Peer review of "Ensembles of decision trees and gradient-based learning for employee turnover rate prediction"

_PeerJ Computer Science, doi:10.7717/peerj-cs.2387_

## Round 0.1 · original submission · Major Revisions

Please incorporate the feedback provided by the two reviewers and make the necessary revisions to the manuscript.

**Language Note:** PeerJ staff have identified that the English language needs to be improved. When you prepare your next revision, please either (i) have a colleague who is proficient in English and familiar with the subject matter review your manuscript, or (ii) contact a professional editing service to review your manuscript. PeerJ can provide language editing services - you can contact us at [email protected] for pricing (be sure to provide your manuscript number and title). – PeerJ Staff

Reviewer 1 ·

Basic reporting

The authors proposed using decision tree ensembles and gradient-based learning to predict employee turnover rates. The problem was solved using one of the new advanced computational methods that were recently published. The manuscript was presented in a professional English style. The problem and research question were clearly defined. The writing structure works well with simple organization. Aside from the novelty of the method used, this work adds value because novel methods are used to address existing popular issues. However, there are some ambiguities in this work that must be resolved (please see below).

Experimental design

(a) This work demonstrates that the authors appear to only implement basic machine learning algorithms. The machine learning algorithms can be used as baseline models for benchmarking, but the comparison was insufficient to conclude that your method outperformed the others. I strongly advise authors to use more advanced methods to compare with their model.
(b) A flowchart for the entire experimental design should be included to provide detailed information on how the models are developed and evaluated.
(c) The learning principle underlying applied methods is unclear. Please create a figure to better explain the algorithm.

Validity of the findings

The authors repeated the experiments 20 times to provide statistical evidence of model reproducibility as well as information on the variation of model performance. The experimental results show that their work has a low variability across multiple random samples.

Reviewer 2 ·

Basic reporting

The goal of this study is to predict employee turnover rates using decision tree ensembles and gradient-based learning. The language used in the manuscript meets the journal's requirements. The organization of all sections was straightforward and easy to follow. The author's work is interesting, with positive outcomes. This work can be published if the authors can address a number of issues regarding the experimental design and results.

Experimental design

- “Gradient-based decision tree algorithm enhances traditional tree-based algorithms by incorporating the principles of gradient boosting” => This sentence is too confusing. Could you please explain how the gradient-based decision tree algorithm differs from the normal tree algorithm?
- Before moving on to the gradient-based algorithm, a paragraph introducing the normal decision tree algorithm is required to provide readers with a general understanding of "what is a decision tree".
- The comparative analysis performed in this work is inadequate because all implemented models use conventional machine learning algorithms. Please compare your model to more advanced approaches, for example, a deep learning-based method.
- Please discuss the limitations of your work and clarify the applicability domain.

Validity of the findings

- A table comparing the features of gradient-based and normal decision trees should be provided.
- Overall, the statistical evidence is complete with minimal variation. In addition to the mean and standard deviation, a 95% confidence interval can be added.

Additional comments

The manuscript is not yet ready for publication at this point.

---

## Round 0.2 · accepted · Accept

The authors have addressed all of the reviewers' comments. The manuscript is ready for publication.

Reviewer 1 ·

Basic reporting

The basic reporting is satisfactory and meets the journal's requirements. The language and structure have been improved in the revised version.

Experimental design

The experimental design has been enhanced with the addition of more comparative algorithms. New figures have been included to provide more details on the method and experiments.

Validity of the findings

Looks OK, no further comments.

Additional comments

The manuscript has been improved. It is now suitable for publication.

Reviewer 2 ·

Basic reporting

No comment

Experimental design

No comment

Validity of the findings

No comment

Additional comments

The authors have addressed all my comments and improved the paper. I have no more comments.